# Initial Development of an Immediate Implantation Model in Rats and Assessing the Prognostic Impact of Periodontitis on Immediate Implantation

**DOI:** 10.3390/bioengineering10080896

**Published:** 2023-07-28

**Authors:** Yingying Wang, Ximeng Cao, Yingyi Shen, Qi Zhong, Yujie Huang, Yifan Zhang, Shaohai Wang, Chun Xu

**Affiliations:** 1Department of Prosthodontics, Shanghai Ninth People’s Hospital, Shanghai Jiao Tong University School of Medicine, No. 639 Zhizaoju Road, Shanghai 200011, China; 2College of Stomatology, Shanghai Jiao Tong University, No. 639 Zhizaoju Road, Shanghai 200011, China; 3National Center for Stomatology and National Clinical Research Center for Oral Diseases, No. 639 Zhizaoju Road, Shanghai 200011, China; 4Shanghai Key Laboratory of Stomatology & Shanghai Research Institute of Stomatology, No. 639 Zhizaoju Road, Shanghai 200011, China; 5Department of Stomatology, Shanghai East Hospital, Tongji University School of Medicine, No. 150 Jimo Road, Shanghai 200120, China

**Keywords:** immediate implantation, animal model, periodontitis, rinse solution

## Abstract

Background: To establish an immediate implantation rat model and to evaluate the effects of pre-existing periodontitis and two different socket rinse solutions on immediate implantation prognosis. Methods: Sprague-Dawley (SD) rats were randomly divided into three groups before immediate implantation, including the control group, the group with experimentally induced periodontitis (EP), in which rats have been experimentally induced periodontitis before implantation, and the group with induced periodontitis and with extraction sockets rinsed with three percent H_2_O_2_ (EP-H_2_O_2_), in which rats have been induced periodontitis before implantation, and extraction sockets were rinsed with three percent H_2_O_2_. Periodontitis was induced by ligating the thread around the molars for four weeks. Six weeks after titanium alloy implants were self-tapped and left to heal transmucosally, maxillae were dissected after the clinical examination to perform micro-CT and histological analysis. Results: An immediate implantation model was successfully built in rats. There was no significant difference in implant survival rates between the EP and control groups. However, the clinical examination results, micro-CT analysis, and histological analysis in EP and EP-H_2_O_2_ groups showed a significantly worse prognosis than in the control group. Three percent H_2_O_2_ showed a similar effect with saline. Conclusion: This study presented a protocol for establishing a rat immediate implantation model and showed that periodontitis history might negatively affect the prognosis of immediate implantation. These findings urge caution and alternative strategies for patients with periodontal disease history, enhancing the long-term success of immediate implantation in dental practice. Additionally, the comparable outcomes between 3% H_2_O_2_ and saline suggest the use of saline as a cost-effective and safer alternative for implant site preparation in dental practice.

## 1. Introduction

The dental implant is a reliable and common treatment for missing teeth [1]. In the 1980s, Per Ingvar Brånemark first introduced the classic protocol called delayed implantation [2]. Delayed implant placement involves a typical three-month waiting period after tooth extraction before placing the dental implant, allowing adequate healing of the extraction site and bone remodeling [3]. However, this protocol has courted controversy, as whether implant placement requires complete socket healing after tooth extraction was unclear. Moreover, this protocol could result in greater burdens for patients as it required extensive clinic visits and overall treatment time [4]. To overcome such limitations, immediate implantation, a protocol that referred to immediately placing the implant in the fresh socket after tooth extraction, was introduced by Schulte and Heimke [5,6], where it was demonstrated to be a superior and effective treatment modality in subsequent studies [7]. Still, there have been many unanswered questions regarding the success of immediate implantation, particularly in biomechanical mechanisms. 

To facilitate a better understanding of immediate implantation mechanisms, the rat model has become an attractive proposition. Rats are commonly used in studies on behavioral phenomena, genetic, biological, and medical purposes due to their short life cycle, adaptability, strong survivability, low cost, and ethical acceptability [8,9]. Since rats share certain physiological similarities with humans in implantation, particularly in bone structure and healing mechanisms, this could provide insights into implant integration and bone responses that are relevant to human conditions [10,11]. However, using rat models in implantology studies has been less of a focus due to structural limitations, making the rodent’s tibia or femur used as implanting sites in previous implant studies [12,13,14,15,16]. However, the anatomy, function, and osteogenesis procedure of the long bone have remained distinctive from those of alveolar bone [17], which could limit direct extrapolation of results to human implantology outcomes. Therefore, the further utilization of rat models would be invaluable in implantology research, providing valuable preliminary insights and contributing to developing safe and effective interventions for human dental implantation. The present study aims to establish a reproducible rat model for research on oral immediate implantation.

Several issues have been indicated as risky factors that might influence implant survival rate, one of which is periodontitis [18]. Under clinical situations, teeth that needed to be extracted due to inflammation were often surrounded by infected tissues. Immediate implantation was usually not proposed if there was inflammation at the implant site, such as the extraction sockets with pre-existing periodontitis [19]. Studies have shown that a history of periodontitis was associated with significant marginal bone loss, higher implant failure rates, and more complications, including peri-implantitis, in the immediate implantation [20,21]. However, some studies have indicated that successful osseointegration could occur when implants were placed at previously infected sites [22,23]. Furthermore, clinical studies were reporting no significant difference in the clinical success of immediate implantation between sites with or without pre-existing periodontitis [24]. Therefore, the present study used the established rat immediate implantation model to evaluate the effects of periodontitis on the prognosis of immediate placed implants.

Different preoperative and postoperative care might have yielded the above inconsistent research results [23]. Several preoperative and postoperative cares are recommended for immediate implantation, including socket debridement, curettage, socket rinse, and systemic antibiotics [23]. Rinsing fresh extraction sockets was necessary to lower bacterial levels in these sites and to expose pristine bone for better healing. In clinical treatments, saline is the most common rinse solution for extraction sockets and periodontal pockets with no cytotoxicity [25]. Three percent H_2_O_2_ is also a commonly used antiseptic and disinfectant agent in dental practice. It is used as an irrigant in dental procedures, including implant surgery and periodontal treatment [26]. Hydrogen peroxide as an irrigant has several potential benefits, including its ability to kill microorganisms, reduce inflammation, and remove debris [27]. However, it is essential to consider the potential drawbacks of H_2_O_2_. Undiluted or high-concentration hydrogen peroxide can cause tissue irritation, mucosal damage, and can be cytotoxic to cells, affecting tissue health and healing [28]. There is still debate regarding the rinsing solutions selection for immediate implantation at the site with pre-existing periodontitis. Therefore, to ensure safe and effective use, the effect of the above two rinsing solutions on immediate implantation was evaluated in the present study. 

On the basis, the hypothesis and null hypothesis of this study are as follows. Hypothesis: The presence of periodontitis history has a negative effect on the prognosis of immediate implantation in rats; the use of 3% H_2_O_2_ for implant site preparation yields similar outcomes to using saline. Null Hypothesis: There is no significant effect of periodontitis history on the prognosis of immediate implantation in rats; there is significant difference in outcomes between using 3% H_2_O_2_ and saline for implant site preparation in rats.

## 2. Materials and Methods

### 2.1. Animals

Seven-week-old male Sprague-Dawley (SD) rats were obtained and then housed under the room temperature at around 20 °C with a light/dark cycle of 12 h. The surgical procedure was pre-approved by the Experimental Animal Ethics Committee of the Ninth People’s Hospital Affiliated to Shanghai Jiao Tong University School of Medicine (relevant judgment’s reference number: SH9H-2022-A84-1).

### 2.2. Group Assignment

Based on Mead’s resource equation, E = (total number of animals) − (number of treatment combination), a good case can be made for E being 25–30 or more to ensure equal group sizes, and it can go even higher when the experimental units are very cheap [29]. In this study the total number of animals is 30, the number of treatment combinations is 2, and E is 28, which is appropriate. The estimated sample size was consistent with those used in previous studies investigating peri-implantitis or periodontitis [15,16,29,30,31,32]. Eight-week-old male SD rats were randomly divided into three groups, 10 in each group, as follows: (a) control group, (b) EP group: with experimentally induced periodontitis for four weeks before immediate implantation surgery, and (c) EP-H_2_O_2_ group: with experimentally induced periodontitis for four weeks before immediate implantation surgery and with extraction sockets rinsed with 3% H_2_O_2_. The description of the group assignment and the whole clinical procedure are shown in Figure 1.

## 3. Induction of Periodontitis

Experimental periodontitis was induced by placing a 2-0 sterile nylon thread ligature around the cervixes of rats’ maxillary second and first molars. The bacteria were continuously inoculated at the thread after ligation (*P. gingivalis* with a concentration of 1 × 10^8^ cfu/mL, inoculated three times with an interval of 30 min daily). To ensure that the thread on the first molar did not fall off, the “figure-of-eight ligation” was used (Figure 2). The periodontitis model was successfully established after four weeks, with reference to the previous report (Figure 2) [33].

## 4. Tooth Extraction and Immediate Implantation

One week before surgery, the rats received daily antibiotics (dose of 100 µL, 20 mg kanamycin, 20 mg ampicillin) through oral infusion [15]. After the rats were anesthetized by intraperitoneal administration of xylazine (5–10 mg/kg body weight) and ketamine (85–90 mg/kg body weight), a specially designed gingival separator was used to sever the soft tissue attachment around their maxillary first molars. Following that, the first molars were extracted by using a specially designed dental elevator and a common mosquito forceps (Figure 3 and Figure 4). For the EP group, sockets were rinsed by saline and for EP-H_2_O_2_ group by 3% H_2_O_2_. Following socket rinsing, a low-speed handpiece was used (drill diameter = 1 mm, 1000 rpm under cooled saline irrigation) to prepare the implant sites in the palatal root area (Figure 4a). A titanium alloy implant (Ti-6Al-4V with anodized surface, BaiorthoTM, Suzhou, China) (Figure 5) was self-tapped and left to heal transmucosally (Figure 6). During the subsequent first week healing period, the rats were administered a daily dose of antibiotics (the same dose before). Their oral health, including rat’s teeth, gums, and oral cavity, was monitored, and overall health, including signs of illness, weight changes, and behavior changes, was checked regularly. At the end of the experiment (6 weeks after implantation surgery), they were humanely euthanized using an overdose of anesthesia. Afterwards, their maxillae were extracted for micro-CT and histological analysis.

All the procedure adhered to the guidelines outlined in the Animal Research: Reporting In Vivo Experiments protocol for conducting experimental studies on animals [34].

## 5. Insertion Torque and Removal Torque Test 

The manual recording of the insertion torque (IT) was performed during the placement of the implant by using an electronic digital torque measuring device (WNS-R-0.5, Weidu Co., Yueqing, China) fitted to the head of the implant. The removal torque value was determined as the maximum rotational force applied in the reverse direction during implant removal until the implant was rotatable horizontally.

## 6. Clinical Examination

Four weeks after tooth extraction and immediate implantation, the rats were anesthetized, and the Mazza bleeding index reflecting the clinical condition of the soft tissue surrounding the implants was assessed using a periodontal probe. The periodontal probe utilized in the study was made of an orthodontic round wire with a diameter of 0.012 inches, with the tip smoothed to prevent injury to the mucosa in the vicinity of the implant. The Mazza bleeding index was clinically assessed using a scale of 0 to 5, as previously documented [29] (Table 1). 

## 7. Micro-CT Analysis

The samples were harvested, fixed, and scanned using a micro-CT (Skyscan 1076, Kontich, Belgium) with a resolution of 9 μm. A volume of interest was defined as the area within a 500 μm radius around the implants to encompass the peri-implant region. The bone tissue evaluation script then generated the final segmented bone image, which included the following parameters: bone volume fraction (BV/TV), trabecular thickness (Tb.Th), trabecular number (Tb.N), trabecular separation (Tb.Sp), bone mineral density (BMD), and bone–implant contact ratio (BIC).

## 8. Histological Analysis

The collected specimens underwent demineralization in 10% ethylene diamine tetracetic acid and were then embedded in paraffin. Thin slices (5 μm thick) were taken from the central part of the samples and stained using the hematoxylin and eosin (H&E) [11]. 

The slices were evaluated in a blinded manner by an experienced histologist under a microscope (CX33, Olympus, Tokyo, Japan). A semi-quantitative histological evaluation was performed to determine the presence or absence of an inflammatory host response. The grading scheme used was adapted from Bleich et al. [35]. The grading methodology was modified to reflect the host immune response in the peri-implant tissues of rats (Table 2). The total score for each slice was calculated by summing up the individual grades.

## 9. Statistical Analysis

The data were represented as mean ± standard deviation and analyzed with SPSS version 26.0 (IBM, Chicago, USA). Differences among the three groups were compared through a one-way ANOVA, LSD test, or chi-square test, and a *p*-value of less than 0.05 was considered statistically significant.

## 10. Results

### 10.1. Survival Rate

The survival rate of implant (implant survival rate = number of final residual implants/number of placed implants) in each group was recorded six weeks post-surgery. One rat died before implantation in Control and EP groups, and two died in EP groups. Therefore, the initial number of placed implants were 18, 18, and 16 in the Control, EP, and EP-H_2_O_2_ groups. The implant number and survival rates of different groups are presented in Table 3. The implant survival rate in control group, EP group, and EP-H_2_O_2_ group is 72.2%, 50%, and 56.3%, respectively. The statistical analysis found no significant difference in implant survival rate among the three groups (*p* > 0.05, chi-square test, Figure 7). 

### 10.2. The Effect of IT on Dental Implant Failure 

According to the IT value, the rats in the control group were divided into three subgroups: 0.5–1, 1–2, and 2–3 Ncm. Results showed that the implant survival rate in 0.5–1 Ncm subgroup was significantly lower than those in 1–2 and 2–3 Ncm subgroups (*p* < 0.05, chi-square test, Figure 8). 

### 10.3. Clinical Examination

The soft tissue condition around the implant was clinically evaluated 4 weeks after the implant was placed. The results indicated that the soft tissue surrounding 84% of the implants (*n* = 18) in the control group was in good condition, as evidenced by their healthy color and texture (Figure 9). In contrast, 50% (*n* = 18) and 44% (*n* = 16) of the implants in EP and EP-H_2_O_2_ groups, respectively, were healthy. It was found that there was substantial bleeding upon probing and gingival swelling only in the EP and EP-H_2_O_2_ groups (Figure 9). The Mazza bleeding index was utilized to determine the extent of probe bleeding, and samples in the EP/EP-H_2_O_2_ group showed significantly higher scores than those in the control group (*p* = 0.022/0.013, one-way ANOVA and LSD test, Figure 10), indicating an inflammatory situation of the peri-implant soft tissues in these two groups.

The soft tissue condition around the implant of the control group was found to be healthy, based on its color and texture, while swelling was present in the EP and EP-H_2_O_2_ groups.

### 10.4. Characterization of the Peri-Implant Osteogenesis

The 3D reconstructions of the dental implants and bone–implant interface from the micro-CT scanning were shown in Figure 11, demonstrating significant differences between EP/EP-H_2_O_2_ and control groups on marginal bone height. There was bone loss at the implant cervical margin area in EP and EP-H_2_O_2_ groups. The control group had a significantly higher value for BIC than the other two groups (EP: *p* < 0.05; EP-H_2_O_2_: *p* < 0.01, one-way ANOVA and LSD test). Additionally, the control group had significantly higher values for BV/TV, Tb.Th, and BMD compared to the EP-H_2_O_2_ group (*p* < 0.05, one-way ANOVA and LSD test). However, no remarkable difference was found in Tb.N and Tb.Sp among the three groups. Results of the removal torque value also showed no difference among these groups (Figure 12).

### 10.5. Characterization of the Peri-Implant H&E Staining

The H&E histological examination evaluated the presence of inflammation in the peri-implant tissue. The histological examination of the control group showed the presence of a normal and clear long junctional epithelium around the implant neck, some with a few neutrophils present, six weeks after the surgery. New bone and osteoid deposits were also observed at the bone–implant contact area, as depicted in Figure 13. Over half of samples of the EP and EP-H_2_O_2_ groups showed inflammatory cells infiltration in the peri-implant tissue, including neutrophil, lymphocyte, plasmacyte, and sometimes together with the appearance of multinuclear giant cells. In addition to migrating inflammatory cells, the EP and EP-H_2_O_2_ groups showed an increase in the number and thickness of collagen fibers in the surrounding soft tissue (Figure 13). The semi-quantitative grading of the host immune response revealed no significant difference in the infiltration of inflammatory cells in the connective tissue and the peri-implant sulcus/pocket between the EP and EP-H_2_O_2_ groups (Figure 14). The fibrosis grading for the control group was higher than for the EP and EP-H_2_O_2_ groups (*p* < 0.05, one-way ANOVA and LSD test, Figure 14). Furthermore, the grading of the bone–implant interface in the control group was found to be higher than that of the EP group (*p* = 0.01, one-way ANOVA and LSD test, Figure 14), resulting in statistically significant differences between the control and EP/EP-H_2_O_2_ groups for these sum scores (Figure 14).

The micrograph of control group showed almost no inflammation, while the inflammatory reaction presented in the EP and EP-H_2_O_2_ groups. In EP and EP-H_2_O_2_ groups, neutrophils migrated into the peri-implant sulcus through the sulcular epithelium. In addition, the number of collagen fibers increased simultaneously and with a higher thickness of soft tissue around the implant in EP and EP-H_2_O_2_ groups (The dotted line indicated the boundary between bone and soft tissue).

## 11. Discussion

The immediate implantation concept was first introduced by Schulte and Heimke in 1976 [6] and the first documented instance of immediate implantation in humans was reported in 1989 [36]. Immediate implantation has become a common treatment in clinical practices [37], with the advantage of minimizing the loss of alveolar bone height, as well as reducing treatment time, compared to the traditional delayed implantation approach which involves waiting for the extraction site to heal before placing the implant [23]. By combining extraction, bone grafting (if necessary), and implant placement into a single appointment, immediate implantation minimizes the number of patients’ visit, eliminates the need for a second surgical intervention, and reduces the overall treatment time and rehabilitation treatment time [38,39]. Although the immediate implantation has gradually become a clinical trend, there are several risks and disadvantages that should be acknowledge. Immediate implantation has shown to be associated with higher risks of post-operative infection and complications, in patients with pre-existing periodontal disease [40]. Achieving initial implant stability can be challenging in immediate implantation cases, particularly when the bone lacks adequate density or volume. Immediate implantation in the anterior region may pose challenges in achieving optimal aesthetic outcomes, as soft tissue contour and healing can be unpredictable [41]. Immediate implantation may lead to mild bone resorption around the implant site during the healing process, which can impact long-term stability and aesthetics. The immediate implantation procedure is also more technically demanding than delayed implantation, requiring precise surgical skills and careful handling of the surrounding tissues. The long-term prognosis of immediate implantation may not be as well-established as that of delayed implants due to the limited availability of long-term clinical data. Immediate implantation’s mechanism, indications, and contraindications are still unclear and need more studies. For such studies, it is necessary to build animal models of immediate implantation. 

Rats are widely used in experimental research because of their moderate size, fast reproduction, short growth cycle, easy feeding, simple administration, appropriate sampling, and diversified behavior [8,9]. As for the dental research, the rat’s oral anatomy and physiology was similar to that of a human [10,11]. The most challenging aspect of establishing a rat model for immediate implantation model was the extraction of the rat’s molar, which contains a small oral cavity and multiple easily broken roots. As Figure 4 presents, an extracted rat first molar has five roots, including three buccal roots and two palatal roots). We demonstrated that, using specially designed instruments and an extraction procedure similar to human first molar extraction, a low root fracture ratio during extraction can be achieved.

The following was to find a suitable implant site. At present, it has been indicated that several factors could influence implant survival rate, one of which was the implant’s primary stability [42]. The objective of attaining good primary stability during implant placement is to reduce excessive micro-movement at the bone–implant interface, thus avoiding fracture of regenerated bone and promoting osseointegration [43,44,45]. According to the principle of good primary stability, this study chose palatal root socket as an implant site, as its size is about 1 mm, which fits the diameter of the implant used in the present study. However, choosing the palatal root socket could be a major challenge. The thin palatal mucosa in rats may be susceptible to wound dehiscence or rupture, leading to postoperative complications. Additionally, the anatomical differences between rat palatal sockets and human alveolar sockets make the direct translation of findings challenging for clinical applications in human dentistry. Regardless, we were able to demonstrate that meticulous surgical techniques and postoperative care are essential to mitigate potential risks and maximize the benefits of this approach in rat studies or preclinical trials.

The IT has been commonly measured in clinical studies to evaluate the primary stability of implants and researchers have also tried to establish a minimum IT requirement for successful implantation [3,14,21,29,30,31,32]. Despite numerous studies and attempts, a clear clinical agreement on the minimum insertion torque value for successful implants has not yet been established. The minimum IT reported by the dental literature ranged from 32–50 Ncm [46,47]. Clinical studies have demonstrated that IT of less than 30 Ncm is associated with early dental implant failures [46,48]. On the other hand, having a high IT during implant placement does not negatively impact the implant’s survival rate or the amount of marginal bone loss [49]. The results of the present study indicate that obtaining primary stability at the time of implant placement is crucial for the successful osseointegration of the implant. Rats have smaller and structurally different jawbones compared to humans. The bone quality, density, and morphology in rat jaws may differ significantly. The diameter, length, and design of implants used in humans and rats are not identical. In human studies, implants are commonly larger and more complex to withstand masticatory forces and provide better primary stability. In contrast, in rat studies, smaller and simpler implants are typically used. These factors could affect the biomechanics of implant placement and the resulting insertional torque values. On this basis, human study results are not directly transferable to rats. In this study, the survival rate of the implant with an IT > 1 Ncm was significantly higher than that with IT < 1 Ncm. This result suggests that a recommended IT for the better application of this immediate implantation rat model should be advised. By doing this, researchers could obtain reliable results in this model without the influence of low IT values. 

Periodontal disease history is another critical factor that might influence the prognosis of immediate implantation. The risk of microbial interference during the healing process has been identified [50]. In clinical practices, it was advised to avoid immediate implantations in cases of periodontal pathosis as it has been linked to an increased risk of peri-implant infections and implant failures as confirmed by several studies [51,52]. Due to this, most clinicians avoid immediate placement of the implants in infected sockets and consider periodontitis a risk to perform immediate implantation [38].

However, some researchers disagree with this opinion. In a study conducted by Marconcini and colleagues, the clinical success of implants placed in fresh extraction sockets with signs of periodontal disease was evaluated [24]. Results showed that all implants were osseointegrated and patients were asymptomatic with no signs of bleeding or infection when probed at the 12 month follow-up [24]. Recent cohort studies reported that the survival rate of implants immediately placed in sites with clinical signs of periodontal disease was comparable to that of implants immediately placed in sites without infection [53,54]. Similar studies performed on dogs have backed up human clinical observations [55,56,57,58].

Based on the established immediate implantation rat model, this study also evaluated the effect of periodontitis on immediate implantation success. The Mazza bleeding index was used to assess rats’ gingival health by evaluating the presence or absence of bleeding from the gingival tissues when gently probed or touched [29]. It provides a quick and easy-to-use method to assess gingival inflammation. Because of rats’ different size and thin gingiva compared with humans, this study chose the Mazza bleeding index, not gingival index or the periodontal index. But Mazza bleeding index is only one aspect of a comprehensive periodontal evaluation. In clinical studies, more comprehensive assessments may include other indices such as the Periodontal Index. Results showed no difference among the three groups on implant survival rates, removal torque value, Tb.N, and Tb.Sp. Nevertheless, there were significant differences among the groups on the results of clinical examination scores, histological analysis scores, BV/TV, BMD, Tb.Th, and BIC. Although results suggested that periodontitis history did not affect the immediate implant survival rate, it might induce a worse prognosis of immediate implantation, as suggested by the results of clinical and histological examinations. Previous research has indicated that more significant marginal bone loss and an increased risk of biological complications such as peri-implantitis in immediate implantation are associated with a history of periodontitis [20,21,59,60]. 

Studies stated that the clinical and histological changes occurred in the peri-implant tissues at the sites with pre-existing periodontitis might be avoided by a valid operative technique if adequate care was taken before and after the surgery procedure [23,24]. The treatment protocol of recent clinical studies about immediate implantation at infected sites included curettage, socket debridement, socket rinse, and systemic antibiotics. In this study, rats were given systemic antibiotics one week before and after implantation to prevent post-operative infections that could compromise the healing process and affect the stability of the implants. The use of antibiotics may influence the overall healing process and the immune response in the animals [61]. This can affect the rate of osseointegration, the dynamics of bone remodeling, and the potential complications associated with implants [62,63]. In the case of confounding effects, the antibiotic type, dosage, and duration were the same in the three groups. Antiseptic solutions to rinse the infected area in the oral and maxillofacial region is a common method to prevent microbial complications [64]. Saline was the most frequently used rinse solution in oral health care and treatment [25]. Other commonly utilized antiseptic rinse solutions included 3% H_2_O_2_, 0.2% chlorhexidine digluconate, 0.5% povidone-iodine, and 0.25% sodium hypochlorite [64]. The present study compared the 3% H_2_O_2_ with saline on the effect on the prognosis of implants immediately placed in the sockets with pre-existing periodontitis and found no difference. However, studies found that prolonged exposure (>1–10 min) to antiseptic solutions could significantly increase the death of bone cells and inhibit growth factor release [64,65]. Therefore, more attention should be paid to socket rinse in experimental protocols and clinical practices for immediate implantation. Experimental studies in dogs have demonstrated that socket debridement and the use of prophylactic antibiotics provide suitable conditions for bone remodeling around implants immediately placed into infected sites [4,18,56,66]. In the EP and EP-H_2_O_2_ groups of the present study, socket rinse and systemic antibiotics were carried out, but socket debridement and curettage were not sufficiently followed because the curette was too big to scrape the socket of the rat thoroughly. This might have resulted in the above changes in the peri-implant soft and bone tissues, but further studies should be conducted to validate this.

In addition to the above-recommended clinical treatment protocols, there were different methods to achieve a better prognosis for immediate implantation at infected sites. Many performed guided bone regeneration procedures [38,39,67,68,69,70]. Some studies included the combination of xenograft, autogenous bone graft, platelet-rich plasma [71], antibiotic solution irrigation [42], peripheral intra-socket ostectomy [38], platelet-rich plasma coating of implant [72], and the use of an erbium laser using photoacoustics to reduce the bacteria in osteotomy sites [69,73]. In future studies, the efficiency of these methods on the prognosis of immediate implantation can be evaluated in this immediate rat implantation model.

## 12. Conclusions

In conclusion, this study successfully built a rat immediate implantation model, and the protocol was described in detail. Based on the research results from this model, the periodontitis history did not influence implant survival rate but might have negative effects on immediate implantation prognosis. Therefore, further studies are needed to clarify the impact of periodontitis. Lastly, 3% H_2_O_2_ as a socket rinse solution showed similar outcomes with saline in the prognosis of the implant immediately placed at the sites with preexisting periodontitis. The comparable outcomes between 3% H_2_O_2_ and saline suggest the possibility of using saline as a cost-effective and safer alternative for implant site preparation in dental practice.

## Figures and Tables

**Figure 1 bioengineering-10-00896-f001:**
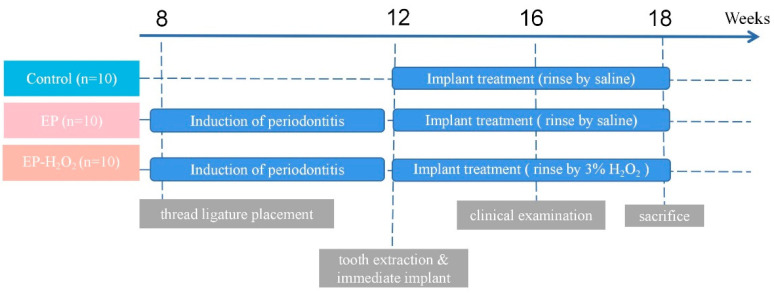
Group assignments and respective treatments. Specific treatments applied at specific time over the establishment procedure of the model were indicated.

**Figure 2 bioengineering-10-00896-f002:**
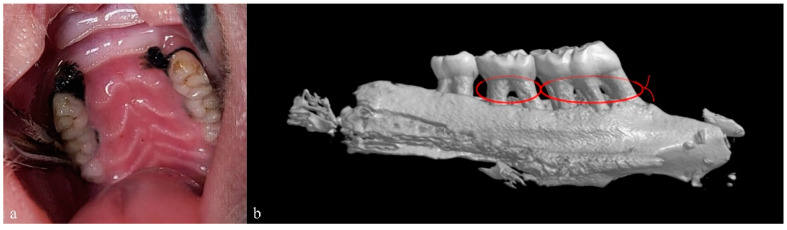
Induction of periodontitis in SD rats. (**a**) The intra-oral photo showing the nylon thread ligation to induct periodontitis in a 12-week-old rat; (**b**) The three-dimensional maxillary reconstruction of the rat from micro-CT scan 4 weeks after ligation, showing the resorption of the alveolar bone due to the inducted periodontitis, and the red line represents the figure-of-eight ligation.

**Figure 3 bioengineering-10-00896-f003:**
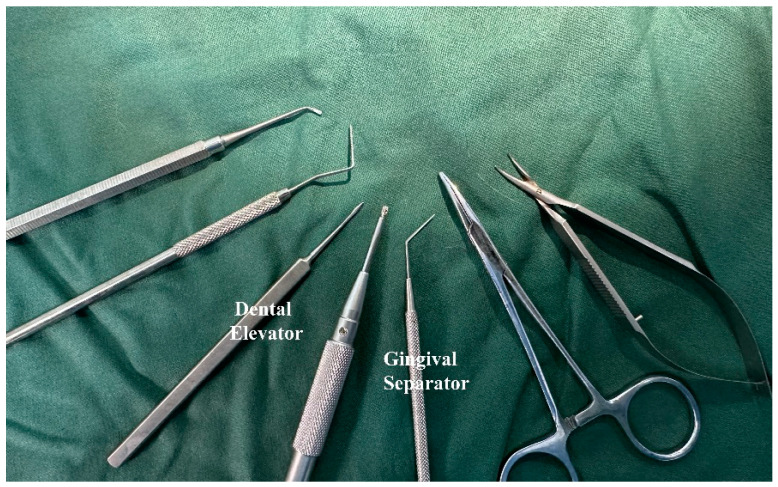
Instruments for tooth extraction and immediate implantation surgery.

**Figure 4 bioengineering-10-00896-f004:**
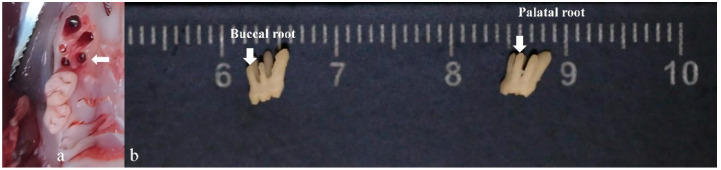
The extraction sockets and the morphology of rat’s first molar. (**a**) The morphology of rat’s first molar extraction sockets. The palatal root socket was used for the immediate implantation site (indicated by the white arrow). (**b**) The morphology of the rat’s first molar. Five roots (three buccal and two palatal were observed (buccal roots and palatal roots are indicated by the white arrows).

**Figure 5 bioengineering-10-00896-f005:**
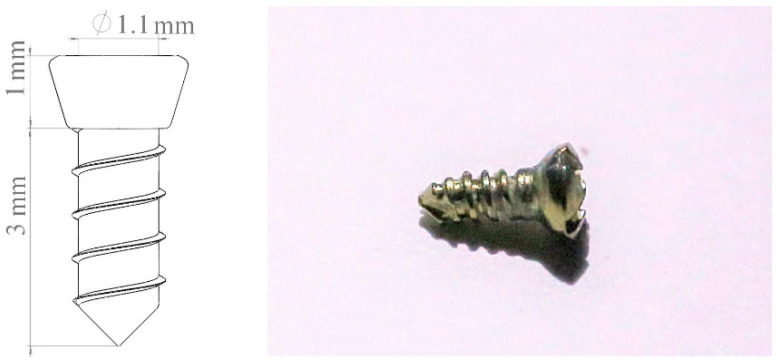
The size and shape of the self-taped implant. The diameter of the implant was 1.1 mm, with a total length of 4 mm and a transmucosal length of 1 mm.

**Figure 6 bioengineering-10-00896-f006:**
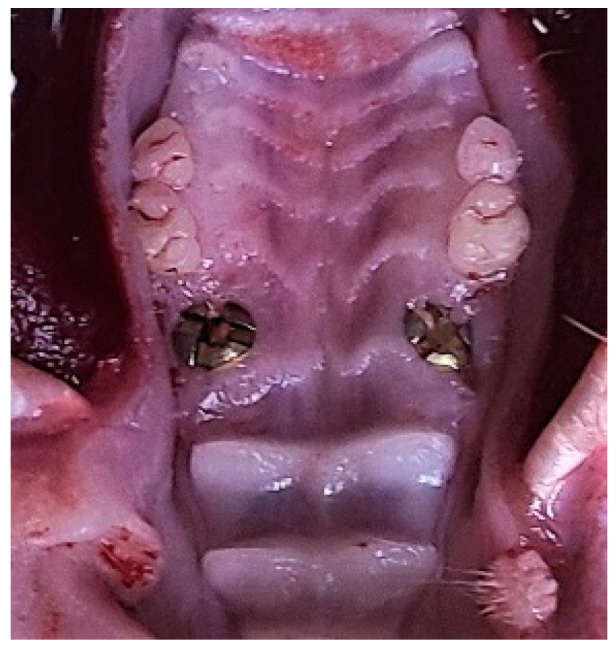
The clinical situation after immediate implantation. Implants were transmucosally placed in the alveolar ridge of the maxillae.

**Figure 7 bioengineering-10-00896-f007:**
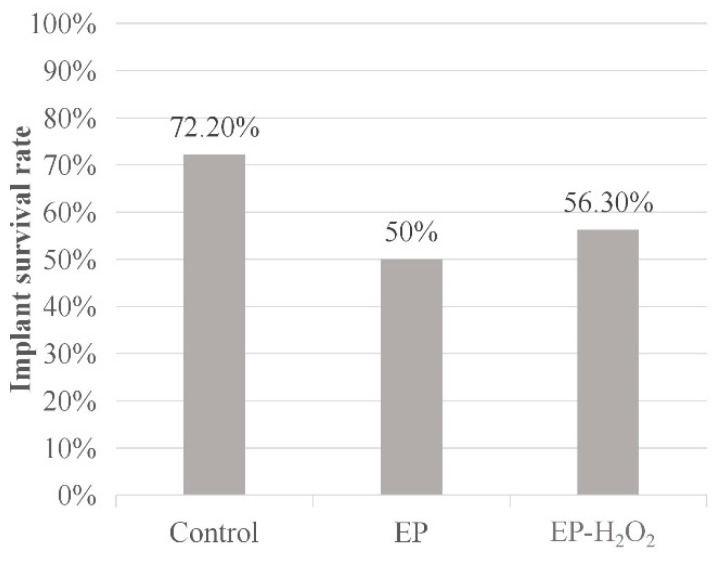
The dental implant survival rates of different groups. The statistical analysis found no significant difference in dental implant survival rate among the groups (*p* > 0.05, chi-square test).

**Figure 8 bioengineering-10-00896-f008:**
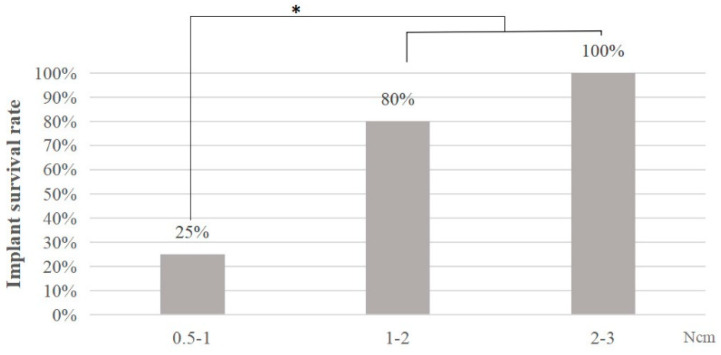
The survival rate of dental implants with different IT values. Statistical analysis results showed that the implant survival rate in 0.5–1 Ncm group was significantly lower than those in 1–2 Ncm and 2–3 Ncm groups (*: *p* < 0.05, chi-square test).

**Figure 9 bioengineering-10-00896-f009:**
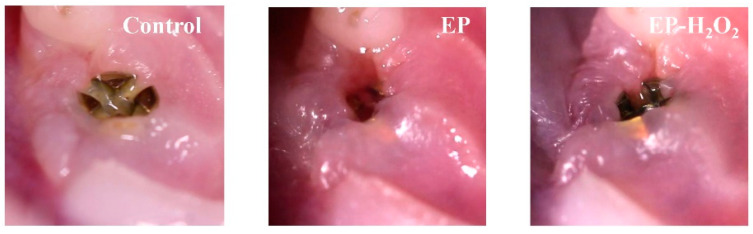
Representative photos of the peri-implant situation.

**Figure 10 bioengineering-10-00896-f010:**
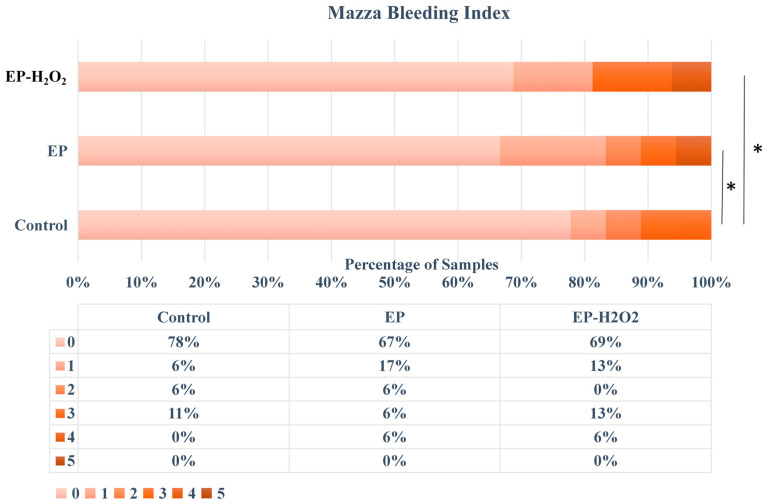
Mazza bleeding index scores of three groups. Samples in EP/EP-H_2_O_2_ group obtained significant higher scores (*: *p* < 0.05, one-way ANOVA and LSD test), indicating the potential inflammatory situation of the peri-implant soft tissues.

**Figure 11 bioengineering-10-00896-f011:**
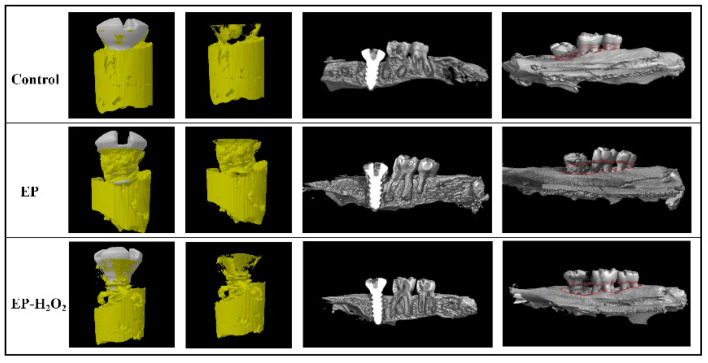
Three-dimensional visualization of the dental implant using Micro-CT technology.

**Figure 12 bioengineering-10-00896-f012:**
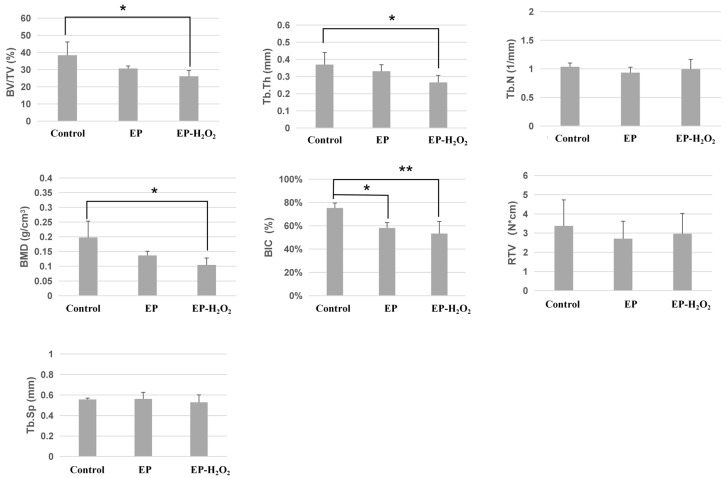
Analysis of BV/TV, Tb.Th, Tb.N, Tb.Sp, BMD, BIC, and removal torque value. The results of the Micro-CT analysis showed that the control group had a higher BIC value compared to the other two groups, and the control group also had a significantly higher value for BV/TV, Tb.Th, and BMD compared to the EP-H_2_O_2_ group. However, there was no significant difference among the three groups for Tb.N, Tb.Sp, and removal torque value (*: *p* < 0.05, **: *p* < 0.01, one-way ANOVA and LSD test).

**Figure 13 bioengineering-10-00896-f013:**
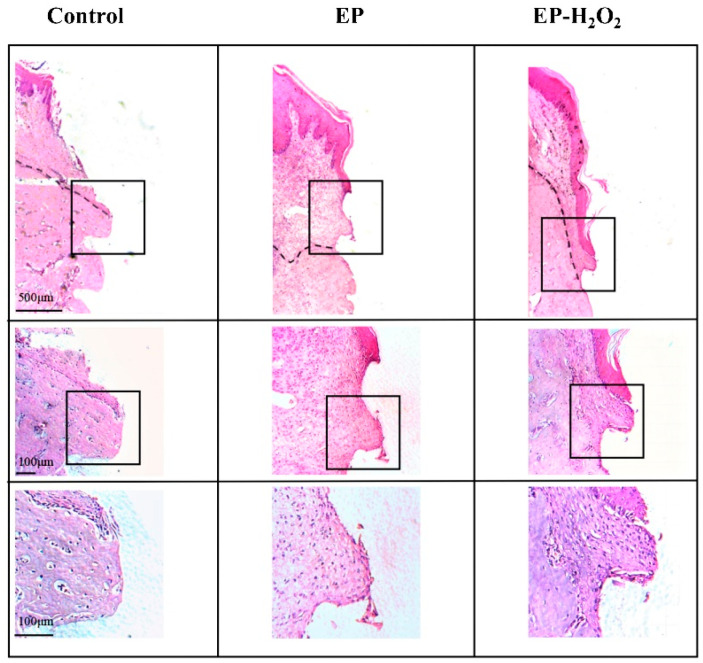
Representative histological micrographs of the peri-implant tissue examined through H&E staining.

**Figure 14 bioengineering-10-00896-f014:**
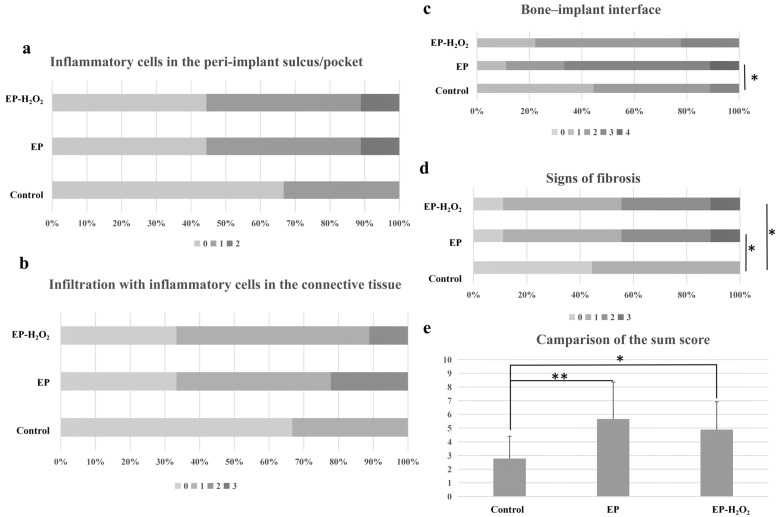
The histological evaluation through a semi-quantitative grading system. (**a**) Inflammatory cells in the peri-implant sulcus/pocket: there was no statistically significant variation among the three groups; (**b**) Inflammatory infiltrations in the connective tissue: there was no statistically significant variation among the three groups; (**c**) Bone–implant interface: the control group received a higher statistical score for the grading in the bone–implant interface compared to the EP group; (**d**) Signs of fibrosis: The grading of fibrosis in control group was statistically higher than EP/EP-H_2_O_2_; (**e**) Sum scores, the differences between control and EP/EP-H_2_O_2_ groups were statistically significant. Significant differences (*: *p* < 0.05; **: *p* < 0.01, one-way ANOVA and LSD test).

**Table 1 bioengineering-10-00896-t001:** Mazza bleeding index.

Mazza Bleeding Index			
Score	Gingiva Looking	BOP	Edema
0	healthy	-	-
1	healthy	+	-
2	unhealthy	+	-
3	unhealthy	+	slight
4	unhealthy	+	obvious
5	unhealthy	+	marked

**Table 2 bioengineering-10-00896-t002:** Semi-quantitative grading system for histological evaluation of infection.

Histological Semi-Quantitative Grading		
Parameter	Score	Historical Appearance
Infiltration with inflammatory cells in the connective tissue	0	No neutrophils/macrophages per field of view
	1	Mild, some neutrophils/macrophages per field of view
	2	Moderate, up to 50 neutrophils/macrophages per field of view
	3	Severe, more than 50 neutrophils/macrophages per field of view
Inflammatory cells in the peri-implant sulcus/pocket	0	No neutrophils
	1	Some neutrophils
	2	Many neutrophils
Signs of fibrosis	0	No
	1	Mild
	2	Moderate
	3	Severe, with prominent neoformation of fibrotic tissue
Bone–implant interface	0	Fully healed (or beginning of healing without signs of bone destruction), mostly osteoblasts
	1	Some osteoclasts/osteoclast activity
	2	Mild osteoclast activity
	3	Moderate osteoclast activity
	4	Severe osteoclast activity (loss of implant)

**Table 3 bioengineering-10-00896-t003:** The implant number and implant survival rates of different groups.

	Control	EP	EP-H_2_O_2_
Number of rats	10	10	10
Number of placed implants	18	18	16
Number of final residual implants	13	9	9
Implant survival rate	72.20%	50%	56.30%

## Data Availability

The datasets used and analyzed during the current study are available from the corresponding author on reasonable request.

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
