# Peer review of "Initial Development of an Immediate Implantation Model in Rats and Assessing the Prognostic Impact of Periodontitis on Immediate Implantation"

_bioengineering, 2023, doi:10.3390/bioengineering10080896_

Round 1
Reviewer 1 Report
The MS is an interesting contribution to animal testing in implantology. However, rats are rodents, genetically and anatomically completely different from humans. Please mention in the Introduction the Pros and Cons of rats in implantology. Please also avoid 95% of unknown abbreviations, in the present form the paper is unreadable! Please also use the 2. Edit. of 2020 Terminologia Anatomica concerning oral orientation buccal ,palatal, anterior and posterior, medial. Do not use the old term mesial. Fig. 11 does not show the maxilla in the view of a mandible.
Use an internationally common reference style, look at the MDPI instructions.
None, minor improvements by specialized translators.
Author Response
Reviewer 1:
The MS is an interesting contribution to animal testing in implantology.
- However, rats are rodents, genetically and anatomically completely different from humans. Please mention in the Introduction the Pros and Cons of rats in implantology.
- Please also avoid 95% of unknown abbreviations, in the present form the paper is unreadable! Please also use the 2. Edit. of 2020 Terminologia Anatomica concerning oral orientation buccal ,palatal, anterior and posterior, medial. Do not use the old term mesial.
- 11 does not show the maxilla in the view of a mandible.
- Use an internationally common reference style, look at the MDPI instruction
Response: The reviewer’s affirmation of our work encourages us to do more studies in this field. We have substantially revised our manuscript according to the reviewer’s comments.
- Rats are commonly used in preclinical research to study implantology and assess the potential efficacy and safety of various interventions before human trials. We have supplemented the information of both the advantages and limitations of using rats as animal models for implantology research in the second paragraph of Introduction.
- The abbreviations EP, EP-H2O2, M1, RTV, EDTA and PRP were replaced by their full names and the changes were highlighted in yellow. The term mesial has been removed, and the description of roots have been changed to three buccal roots and two palatal roots.
- In order to obtain clear image data and avoid interference between the left and right implants, we only took the maxilla and divided it into left and right block sections for shooting micro-CT. Figure 6 showed the implants in the maxillae from the view of mandible.
- We have checked the MDPI instruction as follows, and we have revised the reference style and ensured they consist with the following instruction:
References must be numbered in order of appearance in the text (including citations in tables and legends) and listed individually at the end of the manuscript. We recommend preparing the references with a bibliography software package, such as EndNote to avoid typing mistakes and duplicated references. In the text, reference numbers should be placed in square brackets [ ] and placed before the punctuation; for example [1], [1–3] or [1,3].
Author 1, A.B.; Author 2, C.D. Title of the article. Abbreviated Journal Name Year, Volume, page range.
Reviewer 2 Report
Dear Authors,
Thank you very much for this paper. I think there are linguistic correction needed. The help of an editing service might help.
Title: The title should be rewritten. I recommend a new title: Initial development of an implant.... in rats. Actual it looks like a working title.
Abstract: Please give more scientific information in the abstract. The three groups should be explained in detail. Actually, it is difficult to understand. The used implants should be included. Please weaken your conclusion and add the clinical value of your results for the dental practice.
Introduction:
Please give more information about the different implant strategies. The so-called delayed implant placement is missing. And the information about the healing times in humans should be mentioned in a differential way.
Give more information about the use of H2O2. In many countries and following the clinical guidelines it is discussed in different ways. Please give the best available evidence for the use of H2O2 as irrigant. The role in endodontics is no longer evident. Please remove.
Please give a clear hypothesis and null hypothesis of our study.
Material and Methods:
Please provide more information concerning the sample size calculation. Please clarify.
Figure 3 could be removed. Every dentist knows dental instruments. And the displayed instruments are commonly used.
Figure 4: I can see only three buccal and two palatinal sockets. What do you mean with mesial root? Please clarify.
Give more information about the used implant type and protocol. Did you follow the protocol of the manufacturer are any other protocol.
Line 136: Give exact information about the monitoring of oral and medical health. Please clarify.
Figure should be in higher resolution and bigger size. This might increase readers understanding and overview.
Results:
In most case statistical results were mentioned, but no explained further. Please give more information about the p-values and the test used in these cases. More details of the evaluated data should be provided to follow your results. Please add more information to the figures.
Table 3: Different numbers of implants were placed within the groups. Please explain and discuss.
Discussion:
Please discuss also the disadvantages and risk factors related to immediate implant placement. There are several points, and they should be included.
Please discuss not only the advantages but also the disadvantages by using one the palatinal sockets in rats. There are some disadvantages also.
Discussing the IT: Are the results from human studies transferable on rats. Please discuss. There might be a difference. Also discuss the used diameter and implant design.
Please discuss the use of antibiotics in your model. This remains unclear. There might be the possibility influencing your results. This could be important in interpreting your findings.
Please give more information about the use of the Mazza Index. Why did you use it. There are more common indices.
Conclusion: Following your results there was no difference between the groups regarding survival. Please weaken the conclusion. The impact of periodontitis remains unclear if your results are correct.
Please give a clinical comment or recommendation. What´s about the use of H2O2 if there are no differences compared to saline.
Literature: Most of the references are more the 10-15 years old. Please update the literature to an actual status.
An extensive english revision should be done. Significant improvements are possible.
Author Response
Reviewer 2:
Dear Authors,
Thank you very much for this paper. I think there are linguistic correction needed. The help of an editing service might help.
- Title: The title should be rewritten. I recommend a new title: Initial development of an implant.... in rats. Actual it looks like a working title.
- Abstract: Please give more scientific information in the abstract. The three groups should be explained in detail. Actually, it is difficult to understand. The used implants should be included. Please weaken your conclusion and add the clinical value of your results for the dental practice.
- Introduction: (1) Please give more information about the different implant strategies. The so-called delayed implant placement is missing. And the information about the healing times in humans should be mentioned in a differential way; (2) Give more information about the use of H2O2. In many countries and following the clinical guidelines it is discussed in different ways. Please give the best available evidence for the use of H2O2 as irrigant. The role in endodontics is no longer evident. Please remove; (3) Please give a clear hypothesis and null hypothesis of our study.
- Material and Methods: (1) Please provide more information concerning the sample size calculation. Please clarify; (2) Figure 3 could be removed. Every dentist knows dental instruments. And the displayed instruments are commonly used; (3) Figure 4: I can see only three buccal and two palatinal sockets. What do you mean with mesial root? Please clarifyï¼›Give more information about the used implant type and protocol. Did you follow the protocol of the manufacturer are any other protocol; (4) Line 136: Give exact information about the monitoring of oral and medical health. Please clarify; (5) Figure should be in higher resolution and bigger size. This might increase readers understanding and overview.
- Results: (1) In most case statistical results were mentioned, but no explained further. Please give more information about the p-values and the test used in these cases. More details of the evaluated data should be provided to follow your results. Please add more information to the figures; (2) Table 3: Different numbers of implants were placed within the groups. Please explain and discuss.
- Discussion:(1) Please discuss also the disadvantages and risk factors related to immediate implant placement. There are several points, and they should be included; (2) Please discuss not only the advantages but also the disadvantages by using one the palatinal sockets in rats. There are some disadvantages also; (3) Discussing the IT: Are the results from human studies transferable on rats. Please discuss. There might be a difference. Also discuss the used diameter and implant design; (4) Please discuss the use of antibiotics in your model. This remains unclear. There might be the possibility influencing your results. This could be important in interpreting your findings; (5)Please give more information about the use of the Mazza Index. Why did you use it. There are more common indices.
- Conclusion: (1) Following your results there was no difference between the groups regarding survival. Please weaken the conclusion. The impact of periodontitis remains unclear if your results are correct. (2) Please give a clinical comment or recommendation. What´s about the use of H2O2 if there are no differences compared to saline.
- Literature: Most of the references are more the 10-15 years old. Please update the literature to an actual status.
- An extensive English revision should be done. Significant improvements are possible.
Response: We are very grateful for the reviewer’s careful reading and recognition of our manuscript. The manuscript has been revised substantially according to the reviewer’s comments. We also had the manuscript checked by a colleague fluent in English writing and the linguistic corrections are highlighted in blue.
- Title:
The title has been changed to Initial Development of an Immediate Implantation Model in Rats and Assessing the Prognostic Impact of Periodontitis on Immediate Implantation.
- Abstract:
We have added more detailed information of the three groups and the used implants in Methods of Abstract. The conclusion has been revised to weaken the results and strengthen the clinical value which is highlighted in yellow.
- Introduction:
(1) We have added the information about delayed implantation in the first paragraph of Introduction and the information about the healing times in humans was mentioned after the introduction of delayed implantation.
(2) More detailed information about the advantages and disadvantages of H2O2 as irrigant has been added. It’s role in endodontics has been removed.
(3) A hypothesis and null hypothesis are added in the end of Introduction.
- Material and Methods:
(1) Based on Mead's resource equation, E= (total number of animals) -(number of treatment combination), a good case can be made for E being 25 - 30 or more to ensure equal group sizes, and it can go even higher when the experimental units are very cheap. In this study the total number of animals is 30, the number of treatment combinations is 2 and E is 28 which is appropriate. The above calculation process was explained in the Group assignment of Material and Methods.
(2) Figure 3 shows the instruments used in rat surgery. Because of the rat’s small size, the elevator is essential to the whole procedure's success in our experience of building this model, and the elevator and other instruments are different from those we used in human surgery. Based on the above, we put Figure 3 as a reference for future researchers to do this surgery in rats more effectively.
(3) We have changed it to three buccal and two palatal roots in line 176. The implant we used is the simplest self-taped Ti-6Al-4V implant, as shown in Figure 5. The manufacturer didn’t provide a protocol. The implantation protocol in this study refers to some parts of the protocol in Sun’s study [1] and the protocol in this study is novel.
(4) More specific parameters were given about the monitoring of oral and medical health in line161-163.
(5) Figures were submitted in 1200 dpi. However, in Word, figures might be condensed and so less clear. If the original figures are still unclear enough, we will revise them the first time.
- Results:
(1) Detailed information of the p value and the test used has been supplemented and highlighted in yellow.
(2) Explanations of why different numbers of implants were placed within the three groups is put in the Survival Rate of Results in line 239-241.
- Discussion:
(1,2) Disadvantages and risk factors related to immediate implant placement have been added in line 351-364, and the disadvantages by using one the palatal sockets in rats have been discussed in line 384-390.
(3) The human study results can’t be directly transferable to rats, and we have to supplement the information in line 401-408. Therefore, the results we got about IT in this study are critical for the better application of this immediate implantation rat model. Researchers could get reliable results in this model without the influence of low IT values.
(4) In this study, rats were given systemic antibiotics one week before and after implantation to prevent post-operative infections. In case of confounding effects, the antibiotics type, dosage, and duration were the same in the three groups. The use of antibiotics in implantation has been discussed in line 450-457.
(5) The reason why we chose the Mazza Bleeding Index is discussed in line 429-436. It provides a quick and easy-to-use method to assess gingival inflammation which is suitable for rats’ small size and thin gingiva.
- Conclusion:
(1) The conclusion about the impact of periodontitis has been revised.
(2) Recommendation about the use of H2O2 has been given in the conclusion in line 490-492.
- Literature:
The references have been updated and some old references have been deleted.
- Linguistic correction:.
We have revised the English writing of this manuscript. The revisions have been highlighted in blue.
Round 2
Reviewer 2 Report
Thank you for the revised version of your paper and adressing all comments. I think the paper is now ready for publication. Between both versions significant improvements are present.
The Quality of English writing is improved. Some minor corrections could be done within the final review and editing process.